# Identifying Mental Health Issues in Indian Immigrants in Canada: A Comparison with Non-Indian Immigrants

**DOI:** 10.3390/ijerph22111739

**Published:** 2025-11-18

**Authors:** Sahej Kaur, Mark Rosenberg

**Affiliations:** 1Department of Public Health Sciences, Queen’s University, Kingston, ON K7L 3N6, Canada; 2Department of Geography and Planning, Queen’s University, Kingston, ON K7L 3N9, Canada; mark.rosenberg@queensu.ca

**Keywords:** mental health, immigration, immigrant mental health, Indian immigrants, healthy immigrant effect

## Abstract

Much of the literature on the mental health of immigrants tends to generalize, treating all immigrants as one category, and not accounting for how life experiences in the country of origin can shape mental health. Therefore, the purpose of this study is to contrast the differences in self-rated mental health between Indian immigrants and non-Indian immigrants based on immigration-related factors, sociodemographic factors and health and healthcare utilization-related factors. Cross-sectional data from two cycles of the Canadian Community Health Survey were analyzed. Logistic regression models were analyzed to assess self-reported mental health and those reporting a mood or anxiety disorder. Results provide support for the healthy immigrant effect and find that immigrating in later life is advantageous for mental health for Indian immigrants. Having a lower income, a smaller household, and living in a rural area are associated with good mental health among Indian immigrants, but not among all immigrants. Being male does not have the same protective effect against mental health concerns in Indian immigrants as it does in all immigrants. Results demonstrate the need to study immigrant groups by their country of origin and how life experiences in a particular country shape immigrant mental health differently from country to country.

## 1. Introduction

Research on immigrants and health has often treated all immigrants as a single, aggregate category. This approach ignores the variability in immigrants’ countries of origin and how varying life experiences in the country of origin shape immigrant mental health. Furthermore, it results in the comparison of all immigrants to the general population, classified as non-immigrants, and has led to a recent call for research examining immigrants from individual countries [1]. The reliance on large sample sizes required in many health surveys makes it challenging to collect more detailed information on specific immigrant subgroups in a disaggregated fashion [2]. Thus, quantitative research often examines the healthy immigrant effect and immigrant health by comparing all immigrants with all non-immigrants. This fails to account for differences between immigrants from various countries of origin [3,4].

The types of physical and mental health issues that are most prevalent vary by an immigrant’s country of origin [5]. Therefore, the purpose of this paper is to demonstrate the importance of going beyond aggregate categories of immigrants and non-immigrants to focus on a specific immigrant group and how their mental health status differs from the other immigrant groups. In the research that follows, we contrast Indian immigrants to Canada in comparison to all other immigrants to Canada using the 2015/2016 and 2017/2018 cycles of the Canadian Community Health Survey (CCHS).

In Canada, immigrants comprise 22% of the population [6]. Of all immigrants to Canada, 29% report emotional problems and 16% reported high stress levels [7]. Regarding access to mental health care through outpatient services, emergency department visits and hospitalizations in Ontario, immigrants were 40% less likely than long-term residents to access care [8]. Immigrants may face additional barriers in accessing mental health services and health information, as well as an inadequacy of services that account for differences in cultural and language proficiencies [9,10]. Changes in health upon arrival in Canada may be due to an immigrant’s unique set of challenges, causing the alteration of social, health behaviors and economic experiences [11].

The most common country of origin for immigrants in Canada is India, which has been the case since the 1990s [12]. In 2019, India made up 18.6% of immigrants coming to Canada [6]. More recently, immigrants from India tend to be more educated, skilled and proficient in English [13]. Indian immigrants often must change occupation and industry after immigrating due to a lack of recognition of foreign credentials [13,14].

Research shows that South Asians (including those from India) have unique mental health outcomes compared to other immigrants and non-immigrants in Canada [15]. Compared to non-immigrants, South Asian immigrants display higher rates of anxiety and extremely stressful life events [15]. Stigma towards mental health within the community can act as a significant barrier to seeking treatment for mental health concerns [16,17] and can be associated with immense social consequences such as discrimination and social isolation [18]. There is a lack of research in Canada to identify specific risk factors for mental health unique to the South Asian population, and to more specific groups that fall under the umbrella category of South Asian.

The healthy immigrant effect (HIE) concept relates to the fact that, at the time of immigration, immigrants are, on average, in better health than Canadians [19,20,21,22]. With increasing time in Canada, this advantage decreases, and health deteriorates [20,21,22,23].

Research by Ng & Zhang [24] found that the HIE applies to both self-reported health and mental health. However, research on South Asian immigrants suggests that the HIE lacks nuance and does not fully explain trends in mental health in this group, with first- and second-generation immigrants reporting differing [15]. Similarly, a systematic review of research on the application of the HIE to mental health found large inconsistencies in results, which may be due to variations in the groups included and the lack of sub-group analyses [25]. Thus, there is a research gap when studying how the HIE applies to mental health, how this effect changes over generations of immigrants, and how mental health may differ across various groups of immigrants in Canada [24]. Research on the HIE often does not account for age at the time of immigration [15], and the literature examining the HIE in adolescents is lacking [21].

In addition to the HIE, other factors need to be considered to assess effects on the mental health of Indian immigrants in Canada. Firstly, studying age at the time of immigration is essential as the age composition of immigrants can change due to immigration policies and aging populations. Thus, it is important to consider how the HIE may uniquely impact varying age groups [26]. Secondly, there is a gap in research discussing how living in a rural community impacts the overall health of immigrants [27]. The Mental Health Commission of Canada found that individuals living in smaller communities are more likely to face stigma regarding mental health, which can be especially an issue since it is harder to maintain privacy [28].

Research on the associations between immigration and health is further complicated due to inherent methodological challenges. Canadian immigration prioritizes migration amongst those who are in good health and able to contribute to the economy [29]. Pre-migration experience, including access to education and healthcare, along with the selection of healthier immigrants, acts as a confounder in the relationship between migration and health [30]. In this paper, we address the issues above except pre-migration experience because of the limitations of the CCHS (see below). However, it is likely that the dependent and independent variables used reflect pre-migration experience to some extent.

This study thus addresses the following research question: how do immigration-related, sociodemographic, health, and healthcare utilization factors differ in their associations with (a) good self-rated mental health, (b) reporting a mood disorder and (c) reporting an anxiety disorder between Indian and non-Indian immigrants?

## 2. Data and Methods

### 2.1. Data Source

The analysis below uses data from the 2015/2016 and 2017/2018 cycles of the Canadian Community Health Survey (CCHS). Over each cycle, the entire survey has a sample size of approximately 130,000 individuals. Including both cycles allows for a large enough sample size. In both cycles, data were collected using the same questions, and to the best of our knowledge, no major policy changes occurred that might have affected Indian immigrants or all immigrants to Canada between 2015 and 2018. Access to the CCHS Master Files was gained through an application to the Queen’s University Statistics Canada Research Data Centre (RDC). Under the RDC data sharing agreement, researchers can only gain access to the data in a secure facility, and data cannot be shared with anyone who is not part of the agreement. Only the results of analyses can be released after they have been vetted by Statistics Canada to ensure all results meet protocols on confidentiality. The SAS 9.4 software was used to complete all analyses.

The CCHS is an annual cross-sectional survey disseminated by Statistics Canada. The purpose of the survey is to collect information regarding outcomes such as health status, utilization of health care, determinants of health, lifestyle factors and social conditions. The survey includes individuals over the age of 12 years in all provinces and territories, and data is provided at the health region level. Those who are excluded from the study make up less than 3% of the population of Canadians over the age of 12, and include full-time members of the Canadian Forces, institutionalized populations, individuals living in foster care who are aged 12–17, individuals living in two Quebec health regions and members of First Nations living on-reserve. All questions within the survey are developed by experts at Statistics Canada in conjunction with other academic experts and other federal, provincial and territorial departments. The questions are built to be answered by participants directly using an online questionnaire or on the phone using computer-assisted telephone interviewing software. It is important to note that participation in the survey is voluntary and that interviewers with various language competencies are available to ensure that language is not a barrier. The response for the survey was 58.8% for the 2017–2018 cycle and 59.5% for the 2015–2016 cycle. For more details about the CCHS, see Statistics Canada [31]. Ethical approval was granted by the Queen’s University Health Sciences & Affiliated Teaching Hospitals Research Ethics Board.

### 2.2. Variables

The tables in Appendix A and Appendix B show variables selected from the CCHS used for analysis, with Appendix A showing outcome variables and Appendix B showing exposure variables. The outcome variables selected measure mental health outcomes, including perceived mental health, reporting having a mood disorder or an anxiety disorder. Exposure variables reflect factors related to the mental health of immigrants found to be relevant in the literature review conducted. The CCHS survey measures of perceived mental health, mood disorder, and anxiety disorder are each designed as single-item indicators.

For the age variable, the Statistics Canada departmental standard for age categories based on life cycle groupings was used. The groups used are: children (0 to 14 years), youth (15 to 24 years), adults (25 to 64) and seniors (65 years and over) (Statistics Canada, 2007). These age categories are similar to the categories used by Ng & Zhang [24] for research on immigrant mental health using the CCHS in the past.

The income variable included by the CCHS measures the estimated income per household. Thus, three total household income classes were created for this research: the first included those under the poverty line, and the highest included those with an income of CAD 80,000 or higher. In 2010, Statistics Canada determined that a family of four with a total household income between CAD 35,469 and CAD 38,322 is considered low-income, depending on the location and size of the community [32]. Therefore, the income categories used were CAD 0 to CAD 39,999, CAD 40,000 to CAD 79,999, and CAD 80,000 and more.

For the variable of total household size, data were separated into two groups: those who live in a household of one or two, and those who live in a household of three or more. Due to the sample size, further categories of household size were not feasible. Furthermore, prior research examining household size as a predictor of health outcomes uses the same categorization (e.g., [33]).

For the variable of marital status, data were divided between those in a married or common-law relationship and those who were single, widowed, divorced, or separated.

For the variable of perceived mental health, CCHS asked respondents, “in general would you say your mental health is…?” with responses on a 5-point Likert scale ranging from 1 (poor) to 5 (excellent). A validation study of the use of the CCHS single-item measure of perceived mental health found strong associations with several mental health measures, supporting its effectiveness for assessing the general mental health of a population overall [34]. The use of single-item measures of perceived mental health is commonly used in population-level research (e.g., [35]). Good perceived mental health includes those who reported having good, very good and excellent mental health in the CCHS, and poor mental health included those reporting poor or fair perceived mental health. The dichotomization of the 5-point Likert scale perceived mental health items in this manner is common practice (i.e., [34,35]).

To assess the variables of having a mood disorder and having an anxiety disorder, the CCHS first presents the following information: “Now I’d like to ask about certain long-term health conditions which you may have. We are interested in “long-term conditions” which are expected to last or have already lasted 6 months or more and that have been diagnosed by a health professional.” With regards to anxiety and mood disorders, the following questions are then asked: “Do you have an anxiety disorder such as a phobia, obsessive-compulsive disorder or a panic disorder?” and “Do you have a mood disorder such as depression, bipolar disorder, mania or dysthymia?”. These are answered as yes or no in the CCHS. For researchers interested in how Statistics Canada classifies disorders in more detail, please see the CCHS data dictionary [31].

### 2.3. Data Management and Analysis Strategy

A complete case analysis is used throughout the research. This method was found to be optimal since the distribution of all independent and dependent variables was similar when comparing the cases with one or more missing values, and the cases with no missing values. This indicates that data is missing at random, and that a complete case analysis will not have a large influence on results.

Data analysis was completed in four steps. First, the 2016/2017 and 2018/2019 CCHS datasets were merged. All Canadian-born participants were removed from the merged dataset, and the remaining participants were classified as immigrants from India or immigrants from any other country. The latter group includes immigrants from over 150 origin countries, as defined by Statistics Canada. A complete list of the countries can be accessed by contacting a Statistics Canada RDC. Second, each independent and dependent variable was reviewed to determine whether the response categories required collapsing for analysis. For example, the sex variable included only male or female categories and was left unchanged. On the other hand, the perceived mental health variable was originally measured on a five-point Likert scale and was thus collapsed into two categories. For variables where response categories were collapsed, details of the recoded variables are provided above. Third, contingency tables were created for each independent variable, cross-tabulated with the full set of dependent variables. Chi-square tests were performed to ensure that only independent variables significantly associated with the dependent variables were retained for the regression analysis step. Sensitivity analyses were conducted to confirm that the significant relationship between the independent and dependent variables was not dependent on the grouping of response categories. The fourth step of the analysis was the creation of logistic regression models, as described below. For this paper, we focused on presenting the results of this fourth step. Anyone with questions about steps one to three is invited to contact the corresponding author.

Three logistic regression models were created, each with a different dependent variable: perceived mental health, reporting a mood disorder, and reporting an anxiety disorder. Separate logistic regression models were run for Indian immigrants and non-Indian immigrants for each of the three outcomes, yielding a total of six models. To facilitate comparisons between the two groups, the results for Indian and non-Indian models are presented side by side. Furthermore, the tables are organized by groupings of independent variables (immigration factors, sociodemographic factors, and health and healthcare factors), rather than showing each model in its entirety. Thus, the findings are displayed across nine tables, with each table presenting results for a subset of the independent variables and a single dependent variable for both Indian and non-Indian immigrants. Please see Table 1 s an example of how the results are presented. A method of analysis that directly compares Indian immigrants and non-Indian immigrants in the same model was considered, but ultimately was not used. A stratified analysis, which separates populations into different models, was chosen as it allows us to examine relationships in each population separately and allows comparison of differences between populations.

In all logistic regression models, immigration-related, sociodemographic and health-related variables are included with the following caveat. In some instances, certain variables were not included in some models due to RDC guidelines surrounding data privacy. In these cases, the variable is left blank in the model results table.

Survey weights created by Statistics Canada and included in the CCHS dataset were used in the analyses. Weights are created to ensure that survey data is representative of the entire population, and not just the sample selected. The calculation of the weights is based on the household, using the area of residence and telephone number. Results are applied on an individual level, where the weight given represents the number of people in the population the participant represents within the sample. Analysis was completed with weighted and unweighted data, but only the weighted data are presented in alignment with the guidelines set by Statistics Canada.

For the interpretation of results, the emphasis is on the direction of coefficients compared to the reference group. Although the analyses have produced p-values, we have not relied on statistical significance since the sample is so large and all relationships are statistically significant at the conventional *p* < 0.05, and most at a far lower *p*-value. The only exception was the variable of having a healthcare provider, which was not significantly associated with perceived mental health among non-Indian immigrants.

## 3. Results

### 3.1. Descriptive Statistics

Within the sample of Indian and non-Indian immigrants, there is a roughly even split of women and men. This is representative of the gender diversity found in Indian immigrants and immigrants overall (Table 2). 

There is a smaller proportion of Indians in the 12 to 24 year and 65+ age groups (11% and 14%, respectively) compared to non-Indian immigrants (Table 2). Indian immigrants tend to have higher incomes than immigrants overall, with about 56% in the highest category of CAD 80,000 or more (Table 2). Furthermore, compared to other immigrants and in general, they tend to have larger households of 3 or more individuals (almost 77%) and are more likely to be married (almost 79%) (Table 2). Indian immigrants are less likely than other immigrants overall to live in rural areas, with just less than 4.1% of Indian immigrants living in rural areas (Table 2). Compared to immigrants overall, there is a greater proportion of Indian immigrants who are newer to Canada, having arrived within the past 10 years (42%) (Table 3).

The largest group of Indian immigrants arrived when they were between the ages of 20 to 30 years (about 44%) (Table 3). Indian immigrants reported good perceived mental health and having a mood disorder slightly less often than all immigrants (Table 4). It is important to note that the results of Table 4 include individuals who have comorbid mood and anxiety disorders.

### 3.2. Regression Results

#### 3.2.1. Age at Time of Immigration and Time Since Immigration

In all immigrant groups, immigrating over the age of 31 and being a newer immigrant are associated with advantages for mental health, with this effect being slightly more pronounced in Indian immigrants (Table 5).

We can see that immigrating between the ages of 20 and 30 is associated with an increased odds of mood disorders by more than three times among Indian immigrants (Table 6). 

In both Indian and non-Indian immigrants, a greater time since immigration is correlated with slightly decreased odds of good mental health (Table 5). However, among Indian immigrants, results show increased odds of mood disorders of almost three times after 10 years in Canada (Table 6), which is not seen among non-Indian immigrants. Immigrating between the ages of 20 to 20 reduces the odds of an anxiety disorder in non-Indian immigrants only (Table 7). 

#### 3.2.2. Income, Household Size and Population Size

Results show that income, household size and population size have different effects on the general mental health of Indian immigrants compared to non-Indian immigrants (Table 8). 

Amongst Indian immigrants, having a lower income, a smaller household size and living in a rural area are all factors that increase the odds of good mental health. In contrast, these three factors all slightly reduce the odds of good mental health within non-Indian immigrants, highlighting a discrepancy between the two populations. Although living in a rural area is associated with increased odds of good mental health (Table 8), it is conversely linked to higher odds of reporting a mood disorder (Table 9) among Indian immigrants. 

Rural residence is associated with lower odds of reporting an anxiety disorder (Table 10), consistent with findings showing higher odds of good mental health among Indian immigrants in rural areas.

Among non-Indian immigrants, males have an advantage with respect to all mental health outcomes studied (Table 8, Table 9 and Table 10). This advantage is only slight amongst Indian immigrants and mostly applies to mood (Table 9) and anxiety disorders (Table 10).

#### 3.2.3. Marital Status

Being married increases the likelihood of good mental health among Indian and non-Indian immigrants (Table 8). It decreases the likelihood of reporting a mood disorder (Table 9) or an anxiety disorder (Table 10) to a larger degree amongst Indian immigrants compared to non-Indian immigrants.

#### 3.2.4. Health and Healthcare

As expected, having a high satisfaction with life, low perceived life stress and a strong belonging to the community increase odds of good mental health in both groups (Table 11). 

When looking at access to healthcare, having a regular HCP does not lead to great changes in perceived mental health (Table 11). However, among Indian immigrants, having an HCP decreases the odds of reporting a mood disorder (Table 12) but greatly increases the odds of reporting an anxiety disorder (Table 13). This is the opposite in non-Indian immigrants, with having access to a regular HCP greatly increasing the odds of reporting a mood disorder (Table 12), and only slightly increasing the odds of an anxiety disorder (Table 13).

### 3.3. Summary

Table 14 includes a summary table of the conclusions made among Indian immigrant and non-Indian immigrant models. This provides a simple visual representation of the magnitude and directions of associations between all independent and dependent variables in the two models.

## 4. Discussion

The analysis demonstrates that research aggregating all immigrant groups versus non-immigrants does not accurately reflect patterns within specific immigrant populations. There is a lack of studies focusing on Indian or South Asian immigrants in Canada and their unique risk factors for poor mental health; this research directly addresses that gap.

### 4.1. Immigration-Related Factors Affecting Mental Health

For Indian immigrants who have been in Canada for 10 or more years, there is a 2.8 times increase in odds of reporting a mood disorder. This is a much larger increase than what is seen in the immigrant population overall, and much greater than the decline in general mental health outcomes for Indian immigrants after 10 years. These results provide additional support for the HIE (e.g., [21,24,26,36]). A later age at the time of immigration has roughly equal increases in odds of good mental health among Indian and non-Indian groups. In non-Indian immigrants, it is also associated with lower odds of mood and anxiety disorders. These results align with previous research showing that a later age at the time of immigration is advantageous for mental health (e.g., [37]). However, for Indian immigrants, there is an extremely large increase in odds of a mood disorder for those who immigrate between the ages of 20 and 30 years old. The findings also suggest the need to continue research to identify possible reasons for large increases in mood disorders among Indian immigrants, both among those who are more established in Canada and those who have immigrated between the ages of 20 and 30. Qualitative studies, including interviews or focus groups, and longitudinal research would be useful in understanding the factors driving these patterns.

### 4.2. Sociodemographic Characteristics Affecting Mental Health

Several sociodemographic characteristics have contrasting associations with mental health outcomes in Indian versus non-Indian immigrant populations. The results show that among non-Indian immigrants, males often have a much greater advantage in mental health outcomes. However, this is not the case among Indian immigrants. Males do not have an advantage over females for the general mental health outcome. For self-reports of a mood or anxiety disorder, being male does have a protective effect among Indian immigrants, but this protective effect is relatively much smaller than among the non-Indian immigrant population. Differences related to the culture or lifestyle of Indian immigrants must be further explored to explain mental health differences among Indian men and women. Gender-related analyses are lacking and are necessary to begin to understand these differences [38].

Another way in which the mental health of Indian immigrants deviates from trends seen in other immigrant groups is through income. Among Indian immigrants, a lower income is associated with better self-reported mental health, which is not the case among non-Indian immigrants. This may be due to the fact that for many Indian immigrants, a higher wage will be earned relative to what they were paid in India for lower wage jobs, making poor mental health less likely. However, many recent immigrants from India, China and other Asian nations often do not have their professional qualifications recognized in Canada [39,40]. Furthermore, those who submit a job application with an Indian, Pakistani or Chinese-sounding name are 39% less likely to receive a call back than those with an English name [40]. Individuals who are highly qualified and who are facing these barriers to employment may be forced to work in jobs that do not make full use of their qualifications. Alternatively, they may have to switch jobs many times to reach employment that matches their skillset [14]. This process can lead to feelings of disappointment and frustration [14], which could have a negative impact on mental health.

The association between household size and mental health is also opposite between Indian and non-Indian immigrants, with Indian immigrants in a smaller household having increased odds of good mental health and lower odds of mood disorders, and vice versa for non-Indian immigrants. The presence of family-oriented and collectivist values is integral in South Asian culture [41]. However, this can often lead to unique stressors related to the unjust division of income and resources in traditional larger joint family households [41]. These stressors unique to larger households can lead to poor mental health outcomes, including psychological disorders [41].

Lastly, living in a rural area increases the odds of good mental health in Indian immigrants but not for non-Indian immigrants. However, there are much larger increases in mood disorders for Indian immigrants living in rural areas than amongst non-Indians. The decrease in odds of good mental health seen amongst rural non-Indian immigrants and the increase in odds of mood disorders that rural Indian immigrants face reflect the previous literature finding that rurally based immigrants may face barriers with respect to a lack of social inclusion [42] and less mental health support [43,44]. However, the increase in the odds of good perceived mental health among Indian immigrants in rural areas requires further explanation. This effect aligns with research showing that living in rural areas fosters stronger trust and support networks within Indian immigrant groups [42]. It may also be consistent with evidence suggesting that employment opportunities, particularly for foreign-trained professionals, may be more accessible in rural areas [27]. These results highlight the fact that mental health research focusing on specific immigrant populations is necessary to understand unique mental health risk factors, and that research on mental health for rural immigrant populations is limited in general.

### 4.3. Health Care-Related Factors Affecting Mental Health

Having an HCP is not associated with a large change in odds of good mental health in all groups. It is associated with increased odds of reporting a mood or anxiety disorder, with the exception being the lowered odds of a mood disorder among Indian immigrants with an HCP. Those with mood and anxiety disorders may be the ones who have sought out professional care and have an HCP as a result, or who have been able to better identify their mood or anxiety disorder as a result of having an HCP.

### 4.4. Strengths and Limitations

The main strengths of the research include the large sample size that is representative of immigrants across Canada and the use of high-quality data collection procedures. This includes the use of computer-assisted interviewing and the minimization of non-response through interviewer strategies. Analysis using only Indian immigrants provides a unique perspective, using a large proportion of immigrants to Canada. Furthermore, focusing on age at time of immigration and rural differences has not been heavily researched and provides valuable information on how certain immigrant groups may be disadvantaged with respect to access to care or mental health stressors upon immigration.

There are several limitations to this study that must also be noted. Due to the cross-sectional nature of the survey, only associations between the variables are being examined. No causal inference can be made. Since mental health and overall health can impact whether one immigrates, we cannot ascertain whether poor mental health outcomes occurred before immigration, and whether mental health is associated with immigration itself. Interpretation is limited to potential associations between variables in terms of odds and not in terms of risk, the latter of which might imply temporality. Another important limitation is the use of single-item indicators for the outcomes measured, as this can lack nuance, fail to capture the complexity of the construct being assessed, and does not permit investigation into the underlying causes of poor mental health. Future research using multiple questions would allow for a more focused investigation of specific mental health concerns. Secondly, the use of self-reported diagnoses restricts findings to formally diagnosed cases. This excludes individuals experiencing symptoms with limited access to health services and, thus, a diagnosis. However, at a conceptual level, population health surveys such as the CCHS are not designed to provide the specificity of individual clinical examinations. Their value lies in the ability to identify large-scale trends within populations. For researchers focused on a particular mental health condition, a clinical approach may be more useful, though this method has the opposite limitation in that it is less suited for identifying trends across large populations. Finally, a limitation of this research is the tools and constructs used to identify mental health concerns in immigrant populations. Individuals from different cultural backgrounds may have different definitions and conceptualizations of mental health, which may lead to different interpretations of the questions asked. Furthermore, the mediating role that different forms of discrimination can hold in the relationship between immigration and mental health is not explicitly addressed.

### 4.5. Future Directions and Implications

The results of this study build on previous research on the mental health of immigrants by confirming results related to income, age at time of immigration, the HIE, changes in mental health throughout the lifespan and sociodemographic characteristics associated with positive mental health outcomes. The most notable strength of this research is that it draws attention to major differences that can exist between immigrant groups with respect to mental health. This is evident when looking at how being female, living in a rural area, and having a lower income are associated with poorer mental health in non-Indian immigrant groups, but work in the opposite way for Indian immigrants. Treating immigrant populations as a single, aggregated group can produce results that are not generalizable to specific populations and may create misunderstandings.

Furthermore, the results presented in this paper suggest some areas for future research by identifying factors that are associated with the mental health of Indian immigrants, in contrast with other immigrants. Future research should aim to better understand how these factors influence mental health while accounting for country of origin. Findings from such research can inform the creation of mental health resources, guide resource allocation and healthcare planning for immigrants at greater risk, and enable interventions that address the unique risk factors specific to immigrant populations.

## 5. Conclusions

This study highlights how immigration-related factors, sociodemographic characteristics and health care–related factors are associated differently with mental health when comparing Indian and non-Indian immigrants. These findings underscore the importance of disaggregating immigrant populations in mental health research, as determinants of mental health can vary greatly by country of origin. Distinguishing between immigrant populations is essential to identify unique factors influencing mental health, which can then inform policy and targeted mental health resources to promote positive outcomes for diverse immigrant communities.

## Figures and Tables

**Table 1 ijerph-22-01739-t001:** Modelling strategy.

	Dependent Variables		
Independent Variables	Perceived Mental Health (Model 1)	Mood Disorders (Model 2)	Anxiety (Model 3)
Immigration Factors	Table 5	Table 6	Table 7
Sociodemographic Factors	Table 8	Table 9	Table 10
Health and Healthcare Factors	Table 11	Table 12	Table 13

**Table 2 ijerph-22-01739-t002:** Sociodemographic characteristics of Indian and non-Indian participants.

Sociodemographic Characteristics	Indian Immigrants		Non-Indian Immigrants	
	n	%	n	%
Sex				
Male	898,265	50.6	8,405,891	49.0
Female	876,859	49.4	8,753,723	51.0
Age				
12 to 24 years	199,128	11.2	1,798,704	10.5
25 to 64 years	1,319,256	74.3	12,009,299	70.0
65+ years	256,739	14.5	3,351,611	19.5
Household income				
CAD 0 to 39,999	229,947	12.9	3,873,736	22.6
CAD 40 k to 79,999	550,602	31.0	5,010,694	29.2
CAD 80 k+	994,576	56.0	8,275,184	48.2
Household size				
1 to 2	416,003	23.4	7,010,901	40.8
3+	1,359,121	76.6	10,148,713	59.2
Population Size				
Rural or small	72,253	4.1	1,711,729	10.0
Medium or large	1,702,871	95.9	15,447,884	90.0
Marital status				
Married	1,399,150	78.8	11,446,339	66.7
Single	375,973	21.2	5,713,275	33.3

Source: CCHS (combined datasets from Cycles 2015/2016 and 2017/2018).

**Table 3 ijerph-22-01739-t003:** Immigration-related characteristics of Indian and non-Indian participants.

Immigrant Characteristics	Indian Immigrants		Non-Indian Immigrants	
	n	%	n	%
Country of birth income				
Low income			704,026	4.1
Lower-middle income	1,775,124	100.0	5,349,629	31.2
Upper-middle income			4,423,064	25.8
High income			6,682,895	38.9
Time since immigration				
0 to 10 years	741,604	41.8	5,154,697	30.0
10+ years	1,033,520	58.2	12,004,917	70.0
Age at time of immigration				
0 to 19 years	372,537	21.0	6,198,835	36.1
20 to 30 years	787,066	44.3	5,251,531	30.6
31+ years	615,521	34.7	5,709,247	33.3

Source: CCHS (combined datasets from Cycles 2015/2016 and 2017/2018).

**Table 4 ijerph-22-01739-t004:** Mental health-related characteristics of Indian and non-Indian participants.

Mental Health Characteristics	Indian Immigrants		Non-Indian Immigrants	
	n	%	n	%
Perceived mental health				
Good	1,234,245	69.5	12,451,285	72.6
Poor	540,879	30.5	4,708,329	27.4
Has a mood disorder				
Yes	66,554	3.7	848,887	4.9
No	1,708,569	96.3	16,310,727	95.1
Has an anxiety disorder				
Yes	77,942	4.4	674,814	3.9
No	1,697,182	95.6	16,484,800	96.1

Source: CCHS (combined datasets from Cycles 2015/2016 and 2017/2018).

**Table 5 ijerph-22-01739-t005:** Logistic regression results for immigration-related exposure variables with the outcome of good perceived mental health status in Indian and non-Indian immigrants.

Immigrant Characteristics	Indian Immigrants	Non-Indian Immigrants
Time since immigration		
0 to 10 years (reference category)	1	1
10+ years	0.89	0.96
	(0.88, 0.90)	(0.96, 0.97)
Age at time of immigration		
0 to 19 years (reference category)	1	1
20 to 30 years	1.10	1.20
	(1.09, 1.12)	(1.20, 1.21)
31+ years	1.36	1.28
	(1.35, 1.38)	(1.28, 1.29)

Source: CCHS (combined datasets from Cycles 2015/2016 and 2017/2018).

**Table 6 ijerph-22-01739-t006:** Logistic regression results for immigration-related exposure variables with the outcome of reporting a mood disorder in Indian and non-Indian immigrants.

Immigrant Characteristics	Indian Immigrants OR (95% CI)	Non-Indian Immigrants OR (95% CI)
Time since immigration		
0 to 10 years (reference category)	1	1
10+ years	2.80	1.05
	(2.74, 2.86)	(0.98, 0.99)
Age at time of immigration		
0 to 19 years (reference category)	1	1
20 to 30 years	3.54	0.62
	(3.45, 3.64)	(0.62, 0.63)
31+ years	1.38	0.63
	(1.34, 1.42)	(0.63, 0.64)

Source: CCHS (combined datasets from cycles 2015/2016 and 2017/2018).

**Table 7 ijerph-22-01739-t007:** Logistic regression results for immigration-related exposure variables with the outcome of reporting an anxiety disorder in Indian and non-Indian immigrants.

Immigrant Characteristics	Indian Immigrants OR (95% CI)	Non-Indian Immigrants OR (95% CI)
Time since immigration		
0 to 10 years (reference category)	1	1
10+ years	1.00	1.07
	(0.94, 0.98)	(1.06, 1.02)
Age at time of immigration		
0 to 19 years (reference category)	1	1
20 to 30 years	1.03	0.52
	(1.01, 1.06)	(0.52, 0.53)
31+ years	0.46	0.50
	(0.45, 0.47)	(0.50, 0.50)

Source: CCHS (combined datasets from Cycles 2015/2016 and 2017/2018).

**Table 8 ijerph-22-01739-t008:** Logistic regression results for sociodemographic variables with the outcome of good perceived mental health in Indian and non-Indian immigrants.

Sociodemographic Characteristics	Indian Immigrants OR (95% CI)	Non-Indian Immigrants OR (95% CI)
Sex		
Male	1.01	1.22
	(1.00, 1.02)	(1.22, 1.22)
Female (reference category)	1	1
Age		
12 to 24 years		

25 to 64 years		

65+ years (reference category)		
Household income		
CAD 0 to 39,999	1.28	0.84
	(1.26, 1.30)	(0.84, 0.85)
CAD 40 k to 79,999	0.94	0.95
	(0.94, 0.95)	(0.95, 0.96)
CAD 80 k+ (reference category)	1	1
Household size		
1 to 2	1.36	0.97
	(1.35, 1.37)	(0.97, 0.97)
3+ (reference category)	1	1
Population Size		
Rural or small	1.13	0.94
	(1.11, 1.15)	(0.93, 0.94)
Medium or large (reference category)	1	1
Marital status		
Married	1.08	1.09
	(1.07, 1.09)	(1.09, 1.09)
Single (reference category)	1	1

Source: CCHS (combined datasets from Cycles 2015/2016 and 2017/2018).

**Table 9 ijerph-22-01739-t009:** Logistic regression results for sociodemographic variables with the outcome of reporting a mood disorder in Indian and non-Indian immigrants.

Sociodemographic Characteristics	Indian Immigrants OR (95% CI)	Non-Indian Immigrants OR (95% CI)
Sex		
Male	0.92	0.71
	(0.91, 0.94)	(0.71, 0.71)
Female (reference category)	1	1
Age		
12 to 24 years		

25 to 64 years		

65+ years (reference category)		
Household income		
CAD 0 to 39,999	1.86	1.67
	(1.81, 1.90)	(1.66, 1.68)
CAD 40 k to 79,999	1.24	0.91
	(1.21, 1.26)	(0.91, 0.92)
CAD 80 k+ (reference category)	1	1
Household size		
1 to 2	0.74	1.38
	(0.73, 0.76)	(1.37, 1.39)
3+ (reference category)	1	1
Population Size		
Rural or small	1.89	1.24
	(1.82, 1.97)	(1.24, 1.25)
Medium or large (reference category)	1	1
Marital status		
Married	0.54	0.86
	(0.53, 0.55)	(0.86, 0.87)
Single (reference category)	1	1

Source: CCHS (combined datasets from Cycles 2015/2016 and 2017/2018).

**Table 10 ijerph-22-01739-t010:** Logistic regression results for sociodemographic variables with the outcome of reporting an anxiety disorder in Indian and non-Indian immigrants.

Sociodemographic Characteristics	Indian Immigrants OR (95% CI)	Non-Indian Immigrants OR (95% CI)
Sex		
Male	0.81	0.59
	(0.79, 0.82)	(0.59, 0.59)
Female (reference category)	1	1
Age		
12 to 24 years		

25 to 64 years		

65+ years (reference category)		
Household income		
CAD 0 to 39,999	0.83	1.31
	(0.81, 0.85)	(1.30, 1.32)
CAD 40 k to 79,999	1.28	0.92
	(1.26, 1.30)	(0.92, 0.93)
CAD 80 k+ (reference category)	1	1
Household size		
1 to 2	1.04	1.29
	(1.02, 1.06)	(1.28, 1.29)
3+ (reference category)	1	1
Population Size		
Rural or small	0.46	1.55
	(0.44, 0.49)	(1.54, 1.56)
Medium or large (reference category)	1	1
Marital status		
Married	0.51	0.75
	(0.50, 0.52)	(0.75, 0.76)
Single (reference category)	1	1

Source: CCHS (combined datasets from Cycles 2015/2016 and 2017/2018).3.2.3. Sex.

**Table 11 ijerph-22-01739-t011:** Logistic regression results for health-related variables with the outcome of good perceived mental health in Indian and non-Indian immigrants.

Health-Related Characteristics	Indian Immigrants OR (95% CI)	Non-Indian Immigrants OR (95% CI)
Perceived health		
Excellent or good	5.49	5.08
	(5.45, 5.54)	(5.06, 5.09)
Fair or poor (reference category)	1	1
Satisfaction with life in general		
Satisfied	2.80	3.40
	(2.77, 2.82)	(3.39, 3.41)
Dissatisfied (reference category)	1	1
Perceived life stress		
Some or no stress	1.76	1.58
	(1.75, 1.78)	(1.58, 1.59)
Little or extreme stress (reference category)	1	1
Sense of belonging to community		
Strong	1.30	1.46
	(1.29, 1.32)	(1.46, 1.47)
Weak (reference category)	1	1
Access to regular HCP		
Yes	0.91	1.00 *
	(0.90, 0.93)	(0.99, 1.00)
No (reference category)	1	1
Access to regular HCP—psychologist		
Yes		

No (reference category)		

* Not significant using a threshold of *p* < 0.05. Source: CCHS (combined datasets from Cycles 2015/2016 and 2017/2018).

**Table 12 ijerph-22-01739-t012:** Logistic regression results for health-related variables with the outcome of reporting a mood disorder in Indian and non-Indian immigrants.

Health-Related Characteristics	Indian Immigrants OR (95% CI)	Non-Indian Immigrants OR (95% CI)
Perceived health		
Excellent or good	0.28	0.38
	(0.27, 0.28)	(0.38, 0.39)
Fair or poor (reference category)	1	1
Satisfaction with life in general		
Satisfied	0.32	0.44
	(0.31, 0.32)	(0.44, 0.45)
Dissatisfied (reference category)	1	1
Perceived life stress		
Some or no stress	0.23	0.46
	(0.23, 0.24)	(0.45, 0.46)
Little or extreme stress (reference category)	1	1
Sense of belonging to community		
Strong	0.46	0.65
	(0.46, 0.47)	(0.65, 0.65)
Weak (reference category)	1	1
Access to regular HCP		
Yes	0.76	1.67
	(0.74, 0.78)	(1.66, 1.69)
No (reference category)	1	1
Access to regular HCP—psychologist		
Yes		

No (reference category)		

Source: CCHS (combined datasets from Cycles 2015/2016 and 2017/2018).

**Table 13 ijerph-22-01739-t013:** Logistic regression results for health-related variables with the outcome of reporting an anxiety disorder in Indian and non-Indian immigrants.

Immigrant Characteristics	Indian Immigrants OR (95% CI)	Non-Indian Immigrants OR (95% CI)
Perceived health		
Excellent or good	0.55	0.39
	(0.54, 0.56)	(0.39, 0.39)
Fair or poor (reference category)	1	1
Satisfaction with life in general		
Satisfied	0.41	0.57
	(0.40, 0.42)	(0.56, 0.57)
Dissatisfied (reference category)	1	1
Perceived life stress		
Some or no stress	0.38	0.42
	(0.37, 0.38)	(0.42, 0.42)
Little or extreme stress (reference category)	1	1
Sense of belonging to community		
Strong	0.65	0.94
	(0.64, 0.66)	(0.93, 0.94)
Weak (reference category)	1	1
Access to regular HCP		
Yes	2.36	1.22
	(2.30, 2.42)	(1.21, 1.23)
No (reference category)	1	1
Access to regular HCP—psychologist		
Yes		

No (reference category)		

Source: CCHS (combined datasets from Cycles 2015/2016 and 2017/2018).

**Table 14 ijerph-22-01739-t014:** Summary of conclusions based on Indian immigrant and non-Indian immigrant models.

	Good Perceived Mental Health		Has a Mood Disorder		Has an Anxiety Disorder	
Variable	Indian	Non-Ind	Indian	Non-Ind	Indian	Non-Ind
Immigration-Related						
10+ years since immigration	−1	0	3	0	0	0
31+ years of age at immigration	2	2	2	−2	−3	−2
Sociodemographic						
Male	0	1	0	−1	−1	−2
12 to 24 years of age						
25 to 64 years of age						
Low household income	1	−1	3	3	−1	2
Small household size	2	0	−1	0	0	0
Rural/small population size	1	0	3	0	−3	1
Married	0	0	−2	−1	−2	−1
Health-Related						
Excellent/good health	3	3	−3	−3	−2	−3
Satisfied with life	3	3	−3	−3	−3	−2
Some/no life stress	3	3	−3	−3	−3	−3
Strong sense of belonging to community	1	2	−3	−2	−2	0
Access to regular HCP	0	0	−1	3	3	3

Note: Indian = Indian immigrants, Non-Ind = non-Indian immigrants; −3 = OR < 0.50 (strong negative association); −2 = OR 0.50–0.69 (moderate negative association); −1 = OR 0.70–0.89 (slight negative association); 0 = OR 0.90–1.10 (little or no association); 1 = OR 1.11–1.30 (slight positive association); 2 = OR 1.31–1.50 (moderate positive association); 3 = OR > 1.50 (strong positive association)

## Data Availability

Restrictions apply to the availability of these data. Data were obtained from Statistics Canada and are available at their Research Data Centers with the permission of Statistics Canada.

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
