# Peer review of "Identifying Mental Health Issues in Indian Immigrants in Canada: A Comparison with Non-Indian Immigrants"

_ijerph, 2025, doi:10.3390/ijerph22111739_

Round 1

Reviewer 1 Report

Comments and Suggestions for Authors

This is a really interested and very needed study. Minor changes are however needed to ensure clarity and integrity of the findings and interpretations.

Methods: 

(1) provide scientific corroboration in the categorization employed for household size - may include a citation or a few words on why this categorization was selected.

(2) was the question item "mental health" or "health outcomes" --please clarify as those may have different meaning.

(3) the last paragraph of methods indicates lack of reliance on significate in the examination of the results due to large sample size. It may be helpful to include or list relationships that were not statistically significant somewhere in this section or in the section results.

(4) In the section result, it may be helpful to indicate that the gender diversity found is representative of gender diversity found in Indian immigrants and in immigrants overall.

(5) the constrasting results about income, household size and population size are poignant and this needs to be rewritten. They were better explained in the section discussion but to prevent reader's confusion, please rewrite the 2nd and 4th sentence in this section.

(6) under the section immigration related factors affecting mental health in the discussion section, I encourage you to be more specific on the future research needed - it sounds like mostly qualitative and perhaps quantitative studies are needed to further conceptualize the results.

(7) Include a note/title heading for each Appendix rather than just have Appendix A, Appendix B

Author Response

Thank you for your helpful comments. Please see the attachment.

Reviewer 2 Report

Comments and Suggestions for Authors

While this paper addresses an important topic, I have significant methodological and analytic concerns that prevent me from recommending publication.

Methodological Concerns

  • The collapsing of responses for mental health outcomes is problematic. This approach obscures important nuance and adds further limitations to an already restricted measure—a single, self-reported survey question. Greater justification is needed for why this decision was made, and alternative strategies should be considered.

  • The health outcome measures require far more detail. It is unclear how the mood/anxiety disorder questions were worded. For example, you indicate that participants were classified as having a disorder if they had been diagnosed for six months or expected to have the disorder for the next six months. This method is unconventional, confusing, and risks misclassification. The exact wording of all survey questions should be provided. Moreover, the assumption that participants (especially in non-Western contexts) would know what constitutes a “mood disorder” without clinical framing introduces substantial validity concerns.

  • Limiting measurement to only formally diagnosed cases introduces bias, as many individuals with symptoms will not have access to diagnosis. This substantially narrows the findings and should be acknowledged as a major limitation.

Analytic Concerns

  • The statistical results are underdeveloped. You mention using chi-square tests, but the results are not adequately reported. Were assumptions about normality and distribution tested before choosing these analyses? Results of the chi-squares need to be presented.

  • Odds ratios are not consistently or correctly interpreted. In reporting these, you should clearly state whether predictors increased or decreased the likelihood of the outcome. Betas or odds ratios should be included directly in the results section.

  • Tables are excessive and need to be collapsed, simplified, and labeled more clearly for clarity.

  • Given the large sample size, issues of Type I and Type II error should be addressed explicitly as a limitation.

Interpretation Concerns

  • The lack of statistically significant findings raises questions about how the data are being interpreted. At times, the manuscript appears to draw conclusions not fully supported by the results. The interpretation should remain closely tied to the statistical evidence.

Author Response

(The authors gave the same response as above.)

Reviewer 3 Report

Comments and Suggestions for Authors

Author Response

(The authors gave the same response as above.)

Reviewer 4 Report

Comments and Suggestions for Authors

This excellent manuscript examines differences in mental health outcomes between Indian immigrants and other immigrant groups in Canada using data from the Canadian Community Health Survey (2015–2018). The study addresses an important gap in the literature by focusing on a specific immigrant group rather than treating all immigrants as a homogeneous category. The dataset is robust, the statistical approach is clear, and the topic is highly relevant to public health and immigration policy. The paper is very well structured and clearly written, with a thoughtful discussion of implications and limitations.

Abstract: Consider adding one sentence about the cross-sectional design.

Methodology: The use of data from the Canadian Community Health Survey (CCHS) ensures a large and representative sample. The choice of stratified logistic regression analysis is methodologically appropriate for examining group-specific differences. The reliance on self-reported single-item mental health measures limits validity, as discussed. 

Detailed results presentation: The comprehensive tables provide a wealth of information, and the summary Table XIV is a useful tool to navigate the complex findings.

Tables: The number of large tables makes the results section difficult to follow, but are all important. Given the large number and density of regression result tables (V–XIII), it may help readers if you include a brief note directing them to the summary table first.

In summary Table XIV, the variable “White (racial background)” is shown with values for Indian and non-Indian immigrants (“2” and “0”), but this variable is not mentioned elsewhere in the text, methods, or in the variable classification tables (Appendix B).

The age categories are listed under “Sociodemographic Characteristics” in Tables VIII–XIII, but the corresponding odds ratios are entirely blank. This is misleading; if the variable was not included in these models, it should either be removed or clearly explained. The variable “Access to regular HCP – psychologist” appears in several tables but is consistently empty. An explanation of why it is included despite the absence of data would be helpful.

Discussion: The discussion interprets the results in the context of existing literature and offers plausible explanations for the unique findings among Indian immigrants.

Minor typographical and grammatical issues should be corrected (e.g., “serve as a risk or protective factor”; “Statics Canada”). In addition, the age categories are missing from Table XIV and should be added for completeness. Line 146: The tables show ...

Author Response

(The authors gave the same response as above.)

Reviewer 5 Report

Comments and Suggestions for Authors

Comments on the Quality of English Language

The entire manuscript will benefit significantly from grammatical and sentence structure review.

Author Response

(The authors gave the same response as above.)
